# Lysosomal Biology and Function: Modern View of Cellular Debris Bin

**DOI:** 10.3390/cells9051131

**Published:** 2020-05-04

**Authors:** Purvi C. Trivedi, Jordan J. Bartlett, Thomas Pulinilkunnil

**Affiliations:** 1Department of Biochemistry and Molecular Biology, Dalhousie University, Halifax, NS B3H 4H7, Canada; purvi.trivedi@dal.ca (P.C.T.); jjeffreyb@mun.ca (J.J.B.); 2Dalhousie Medicine New Brunswick, Saint John, NB E2L 4L5, Canada

**Keywords:** lysosome, metabolism, autophagy, endocytosis, mannose-6-phosphate, cathepsin, calcium, proton

## Abstract

Lysosomes are the main proteolytic compartments of mammalian cells comprising of a battery of hydrolases. Lysosomes dispose and recycle extracellular or intracellular macromolecules by fusing with endosomes or autophagosomes through specific waste clearance processes such as chaperone-mediated autophagy or microautophagy. The proteolytic end product is transported out of lysosomes via transporters or vesicular membrane trafficking. Recent studies have demonstrated lysosomes as a signaling node which sense, adapt and respond to changes in substrate metabolism to maintain cellular function. Lysosomal dysfunction not only influence pathways mediating membrane trafficking that culminate in the lysosome but also govern metabolic and signaling processes regulating protein sorting and targeting. In this review, we describe the current knowledge of lysosome in influencing sorting and nutrient signaling. We further present a mechanistic overview of intra-lysosomal processes, along with extra-lysosomal processes, governing lysosomal fusion and fission, exocytosis, positioning and membrane contact site formation. This review compiles existing knowledge in the field of lysosomal biology by describing various lysosomal events necessary to maintain cellular homeostasis facilitating development of therapies maintaining lysosomal function.

## 1. Introduction

Seminal studies by Duve Laboratory uncovered lysosome as the cellular compartment for the degradation of biological macromolecules [1,2]. Endocytic [3,4], autophagic [5,6] and phagocytic [7,8] pathways facilitate macromolecule degradation within the lysosome. Acid hydrolases and lysosomal membrane proteins (LMPs) dictate lysosomal function [9,10]. The acidity of the lysosome stabilizes and mediates the activity of ~60 luminal hydrolytic enzymes. The lysosomal limiting membrane harbors ~25 LMPs, which include transporters, trafficking/fusion machinery, ion channels and structural proteins [10]. LMPs are pivotal in maintaining lysosomal membrane integrity, luminal acidification, an ionic gradient and homeostasis, protein translocation and membrane trafficking [9,10]. In addition, lysosomes contain ions and harbor ion channels, which exert an indispensable role in regulating lysosomal pH and function [11].

Beyond the lysosome’s canonical role in cellular waste disposal, it is also implicated in nutrient sensing, immune cell signaling, metabolism, and membrane repair [12]. Emerging studies show that intra-lysosomal and extra-lysosomal processes govern lysosomal fusion and fission [13], exocytosis [14], positioning [15] and formation of a membrane contact site [16]. Lysosome fusion and fission influence lysosome number, size and exocytosis [13,14,15]. Furthermore, depending on cellular metabolic requirements, or activation by distinct stimuli, lysosomes mobilize to either the cell periphery or to the perinuclear region [15]. Lysosomes also form a membrane contact site with other organelles to exchange signaling information, shuttle metabolites and render ionic homeostasis [16,17]. Perturbation in lysosomal function is observed in lysosomal storage disorders, neurodegenerative conditions, cancer, and cardiovascular and metabolic diseases. This review compiles existing knowledge in the field of lysosomal physiology and function by describing lysosomal events necessary in maintaining lysosome function and cellular homeostasis. 

## 2. Lysosome Biogenesis 

Lysosomes are 0.2–0.3 μm in diameter. Primary lysosomes originate from the Golgi apparatus. Current literature describes multiple models of lysosomal biogenesis. The first model of lysosome biogenesis describes the formation of early endosomes (EEs) from the plasma membrane, and their progressive maturation to late endosomes (LEs) and lysosomes [18,19]. The second model involves vesicular transport, where endosomal carrier vesicle/multivesicular bodies (ECV/MVBs) transfer cargo from early to late endosomes to lysosomes or directly from the matured LEs to lysosomes [18,19]. The third model denotes the “kiss and run” event wherein, LEs (“kiss”) form a contact site with lysosomes transferring cargo with ensuing dissociation (“run”) of lysosomes and LEs [18,19]. The fourth model of lysosome biogenesis is purported to be a fusion-fission event involving a heterotypic fusion of LEs-lysosomes to form hybrid organelles, followed by lysosome re-formation (Figure 1). 

### 2.1. Sorting

Lysosomal enzyme precursors are biosynthesized in the rough endoplasmic reticulum and modified in the Golgi apparatus [20]. Newly synthesized enzymes are initially tagged with mannose-6-phosphate residues, targeting them for specific binding to the mannose-6-phosphate receptors (M6PRs) in the trans-Golgi network (TGN). Subsequently, enzymes tagged with M6PRs are packed into plasma membrane localized clathrin-coated vesicles (CCVs) for biosynthetic transport to LEs either directly or indirectly via EEs [21,22]. Sorting begins by budding and fusion of CCVs with each other or with pre-existing EEs to deliver and sort endocytic cargo [10,23]. 

EEs have acidic intraluminal pH, facilitating uncoupling of ligands from M6PRs and allowing migration towards LEs and lysosomes [4]. Unbound M6PRs are either transported back from endosomes-to-TGN via endosomal vacuoles or are recycled via tubular-sorting endosomes (TSEs) [21,22]. This retrograde recovery of M6PRs from endosome-to-TGN occurs via a “retromere”, which is a penta-unit structure that encodes vacuolar protein sorting (VPS) genes [21]. Structurally, a retromere is comprised of a dimer of sorting nexin proteins, which include SNX1/2 (Vps5-Vps17 in yeast) and a trimeric core of Vps35-Vps29-Vps26 referred to as a cargo-selective complex (CSC) [24]. Monomeric G proteins like RAB GTPases, such as Rab5 and Rab7, regulate the association of retromeres with EEs [25]. Loss of Rab7 function destabilizes the trimeric structure of the retromere, disrupting M6PR recycling and acid hydrolase sorting, impairing EE-to-LE maturation and disrupting cargo degradation [26]. Unlike lysosomal hydrolases that bind with M6PRs, LMPs are delivered either directly to endosomes and lysosomes (direct pathway) or indirectly to the plasma membrane and then to endosomes and lysosomes (indirect or salvage pathway) [27]. 

The sorting of acid hydrolases and LMPs requires the heterotetrameric adaptor protein complex AP1, AP2, AP3 and AP4, each composed of four adaptin subunits. Specific localization of AP governs sorting of M6PRs and LMPs and thereby lysosome biogenesis. AP1 is localized to the TGN and endosomes and assists in the recycling of M6PRs within the TGN. AP2 and AP3 are located on the plasma membrane and endosomes respectively, aiding the transportation of LMPs to lysosomes. Loss of AP3 function in Hermansky-Pudlak syndrome 2, causes redistribution of LMPs to the plasma membrane and impairs lysosome biogenesis in melanosomes and platelet dense granules [28,29]. 

### 2.2. Vesicular Transport and Maturation 

Transport from early to late endosomes involves selective sorting events to facilitate cargo movement into the lysosome. Multivesicular endosomes/bodies (MVBs) are endosomes that contain intraluminal vesicles (ILVs) (Figure 1). Progressive accumulation of ILVs and sorting of lysosome-directed proteins onto ILVs characterizes typical MVBs maturation. The cargo proteins destined for degradation are ubiquitinated and recognized by the ESCRT complex (endosomal sorting complexes required for transport), driving the ILV biogenesis necessary for sorting cargo proteins. When multiple rounds of membrane fusion and fission occur, EEs enriched in ILVs are structurally remodeled to form globular LEs and subsequently lysosomes for degradation. Vesicular transport and maturation of EEs to LEs depend on the conversion of Rab5 to Rab7. As EEs progressively mature and acidify into LEs, they move from cell periphery to cell center. During maturation, EEs lose Rab5 on their membrane and gain Rab7, underscoring the importance of Rab proteins in the EE-to-LE maturation process [26].

Additionally, maturation of EEs to LEs/lysosomes requires v-ATPase, a proton pump that acidifies LEs/lysosomes to a pH of ≈5.5/5.0. In addition to proton homeostasis, ionic Ca^2+^ balance within LEs and lysosomes is indispensable for endo-lysosomal functions including receptor-ligand uncoupling and lysosomal enzyme transport and activity. Indeed, homotypic (fusion of early and late endosomes) and heterotypic (fusion of LEs and lysosomes) fusion are dependent on the Ca^2+^-calmodulin complex. This was shown via cell-free experiments, where endosome-lysosome fusion was suppressed by treating cells with either BAPTA (1,2-bis(*o*-aminophenoxy)ethane-N,N,N′,N′-tetraacetic acid; binds to Ca^2+^ 100 times faster) or an ester form of EGTA-AM (ethylene glycol-bis(β-aminoethyl ether)-N,N,N′,N′-tetraacetic acid-Acetoxymethyl ester; AM dissociates within the lumen leading to luminal Ca^2+^ chelation) but not with EGTA (exchanges Ca^2+^ ~100 times slower than BAPTA). 

## 3. Lysosomal Enzymes

Lysosomes are abundant in hydrolytic enzymes such as proteases, sulfatases, nucleases, lipases, phosphatases, glycosidases and nucleases, all of which degrade complex macromolecules. Lysosomal enzymes have optimal activity at pH 5.

### 3.1. Cathepsin Proteases

Discovered in the gastric fluid in 1929, cathepsin literally means to digest or boil down. According to their structure and catalytic activity, cathepsins are classified into serine cathepsins (A and G), aspartic cathepsins (D and E) and cysteine cathepsins. Among the cathepsins, the most highly characterized proteases are cysteine proteases, which consist of 11 cathepsins (B, C/DPP1, F, H, L, K, O, S, V, W and X) (Figure 2) [30,31]. Cathepsin activity is regulated by transmembrane protein presenilin 1 (PS1) that targets the V0a1 subunit of v-ATPase to maintain an intraluminal acidic environment, which is vital for the stability of lysosomal proteases. Similarly, the abelson (Abl) family of cytoplasmic non-receptor tyrosine kinases, including Abl1 and 2, also govern autophagy and lysosome proteolytic activity [32]. Indeed, loss of function of Abl1 in A549 alveolar carcinoma cells leads to suppression of lysosomal enzymatic activity and impairs lysosome localization and motility, leading to the accumulation of autophagosomes [32].

Proteases play an integral role in proteolytic processing, protein modification and degradation of low-density lipoproteins (LDL) [33,34,35]. Human macrophages and smooth muscle cells incubated with lysosomal protease stimulator, zymosan, experience accelerated LDL hydrolysis to apolipoprotein B-100 and triglycerides (TAGs), triggering the formation of foam cells, an effect sensitive to protease inhibitor pepstatin A [36]. Mice deficient in cathepsin B or cathepsin L are resistant to inflammation and atherosclerosis upon cholesterol crystal injection [37,38]. Comparatively, cathepsin G, an angiotensin II-forming serine protease, displays elastolytic activity in mast cells, causing adverse remodeling and progression to aortic stenosis [39,40]. These findings highlight the role of cathepsin proteases in the pathogenesis of atherosclerosis-based valvular diseases (Figure 2). Furthermore, mice with combined deficiency of cathepsin B and L display an accumulation of endo-lysosomes in the brain, leading to brain atrophy and death in the second and fourth week of life [41]. This study highlights the role of lysosomal proteases in central nervous system maturation during the postnatal period. Cathepsins B and D also facilitate receptor-mediated endocytosis of insulin in rat liver [42], whereas cathepsins B and D degrade insulin-like growth factor (IGF) in MG-83 [43] and HepG2 [44] cells. Notably, cathepsin L is involved in the proteolysis of IGF binding proteins (IGFBPs) in mouse dermal fibroblast [45], suggesting that lysosomal cysteine proteases are critical for growth factor signaling by influencing receptor-mediated growth factor degradation. 

Lysosomal proteases also play a significant role in regulating macroautophagy. The ubiquitin-like conjugation system aids in the processing of Atg8, a step crucial for autophagosome formation. In yeast, autophagosomal formation is aided by cysteine protease Atg4, which cleaves the C-terminus of Atg8 (GABARAP, LC3 and GATE-16, mammalian Atg8 homologs). Subsequently, Atg8 conjugates with PE (phosphatidylethanolamine) to form Atg8-PE complex, a process catalyzed by Atg7 and Atg3 [46,47,48]. Aspartic protease cathepsin E assists in lysosomal autophagy by regulating lysosomal luminal pH, independent of its action on v-ATPase activity [49]. Indeed, mouse macrophages deficient in Cathepsin E accumulate lysosomal membrane proteins with increased lysosomal pH [49], phenocopying lysosomal storage disorders. Together, these studies signify that cathepsin proteases are essential for lysosomal autophagy function and help in maintaining lysosomal structural integrity. 

### 3.2. Lysosomal Acid Lipases (LAL) 

Encoded by lipase A (LIPA), LAL plays a critical role in energy metabolism, signaling and structural integrity of cell membranes [50]. LAL is primarily responsible for hydrolyzing lipoprotein cholesteryl esters (CEs) and triglycerides (TAGs) to free cholesterol (FC) and fatty acids (FAs), respectively (Figure 2). LAL hydrolyzes exogenous and endogenous neutral lipids, which are primarily delivered to the lysosome either through endocytosis of lipoproteins or through autophagy (also known as lipophagy) [51,52].

Under physiological conditions, LAL hydrolyzes acetylated-LDL containing CEs into FC, which can be primarily stored as a lipid droplet or used for membrane assembly. On the other hand, oxidized-LDL is a poor substrate for LAL, causing retention of both CEs and FC within the lysosome of foam cells promoting atherogenesis. Lysosomal cholesterol accumulation is associated with the pathogenesis of atherosclerosis [53,54]. Electron microscopy within an atherosclerotic lesion reveals lipid-laden lysosomes both in human lesions [55,56] and in animal models of atherosclerosis [57,58,59]. Importantly, overload of CEs and FC increases lysosomal luminal pH by inhibiting vATPase, resulting in loss of LAL hydrolytic activity [50]. LAL deficiency or inactivity causes Wolman disease (WD), which has infant-onset and is fatal, whereas CE storage disease (CESD) has a later-onset due to residual (5–10%) LAL activity manifested in patients with hyperlipidemia, hepatosplenomegaly and premature atherosclerosis [60,61]. Notably, recent genome-wide association studies have identified coding LIPA variant with altered processing of LAL signal peptide associated with coronary artery disease. Furthermore, macrophages from individuals with the risk-LIPA coding allele exhibit increased degradation of LAL, resulting in reduced LAL expression and LAL activity compared to the non-risk allele [57]. LAL-deficient mice injected with human recombinant LAL (rhLAL) shows reversal of pathogenic lipid accumulation during atherosclerosis [62]. Sebelipase alfa, an rhLAL, is currently FDA approved for the treatment of Wolman disease and CESD [63,64]. However, it is still not clear whether rhLAL therapy is suitable to treat patients with atherosclerosis unrelated to LAL deficiency. T cell proliferation and maturation in the thymus is abolished by inhibition of CEs and TAG metabolism in LAL-deficient mice, triggering organ inflammation and damage [65]. LAL-deficient mice also exhibit substantial accumulation of CEs and TAGs in the liver [66], adrenal glands, and small intestine with progressive loss of both white and brown adipose tissue [67]. Additionally, TAG hydrolysis liberates free FAs, which are essential for M2 macrophage activation and resistance to infection. Conversely, inhibition of LAL-dependent lipolysis suppresses immunity against pathogens [52]. Together, these findings suggest that LAL and its lipid mediators are essential for immune cell maturation and function by playing a key role in cellular differentiation and metabolism. 

### 3.3. Sulfatases 

Sulfatases are an evolutionarily-conserved enzyme family classified into non-lysosomal and lysosomal sulfatases based on their subcellular localization and pH preference [68,69]. Sulfatases catalyze the hydrolysis of glycosaminoglycans, including heparin, heparan sulphate, dermatan and keratan sulphate, sulfolipids (cerebroside-3-sulphate) and sulphated hormones (dehydroepiandrosterone-3-sulphate) (Figure 2), which play a critical role in cellular degradation, signaling and hormone function [68,69]. Sulfatase activation require C_alpha_-formylglycine (FGly) in their catalytic site, and mutations at this site lead to multiple sulfatase deficiencies characterized by severe metabolic and developmental abnormalities [68,70]. While transporting to the final endo-lysosomal compartment, sulfatases become heavily glycosylated in the ER and Golgi, and processed via the secretory pathway [68]. 

### 3.4. Nucleases

Nucleases degrade RNA and DNA delivered to the lysosome via a process known as RNautophagy/DNautophagy (RDA) (Figure 2) [71]. Notably, lysosomal membrane protein LAMP2C facilitates the translocation of nucleic acids to the lysosome [72,73]. LAMP2 knockout mice exhibit impairment in RDA activity, whereas LAMP2C-overexpressed lysosomes display higher RDA activity [72,73]. Furthermore, LAMP2C directly binds to RNA and DNA before translocation to the lysosomal lumen [72,73], where they are degraded via lysosomal nucleases, RNase T2 and DNase II, respectively [74,75]. RNase T2 is a ribonuclease enzyme that cleaves single-stranded RNA into mono- or oligo-nucleotides, whereas the human homolog RNASET2 cleaves poly-A and poly-U oligonucleotides [74,76]. Hey4 cells (ovarian tumor cell line) transfected with RNASET2 demonstrate that RNASET2 is produced as a full-length nuclease in the secretory granules localized at the lysosome and exhibit optimal catalytic activity in acidic pH [74,76]. Likewise, DNase II is an endonuclease enzyme localized in the acidic lumen of the lysosome and is responsible for cleaving double-stranded DNA [75]. 

## 4. Ionic Balance in Lysosome Function

Lysosomal ionic gradient is indispensable for intraluminal acidification, acid hydrolase stability and macromolecule degradation. Lysosomal ions are involved in lysosome biogenesis, motility, membrane contact site formation, and lysosome homeostasis. The lysosomal lumen is comprised of various ions including Ca^2+^, Na^+^, K^+^, Zn^2+^, H^+^ and Fe^2+^ (Figure 2). 

### 4.1. Ca^2+^

The lysosomal luminal concentration of Ca^2+^ is ~0.5 mM, which is 5000-fold higher than the cytosolic Ca^2+^ store. The endosomal and lysosomal Ca^2+^ flux is pH dependent and is essential for signal transduction, organelle homeostasis and acidification [77,78,79]. Ca^2+^/H^+^ exchangers (CHX) purportedly maintain the lysosomal Ca^2+^ gradient, wherein Ca^2+^ is transported into the lysosomal lumen in exchange for the removal of a proton [80]. Ca^2+^ is required for endosome-lysosome fusion, an effect sensitive to Ca^2+^ chelators, such as BAPTA and EGTA-AM [81]. Fused organelles (LEs-lysosomes) treated with EGTA-AM or bafilomycin A1, exhibit reduced lysosome re-formation, suggesting that both luminal lysosomal Ca^2+^ and acidic pH are indispensable for lysosome maturation [82]. Lysosomal Ca^2+^ also regulates lysosomal mobilization, exocytosis, membrane/vesicle trafficking and membrane contact site formation [79].

### 4.2. H^+^

Lysosome function is dependent on luminal acidification, since acidic pH renders lysosomal hydrolases stable and active, facilitating macromolecule degradation and vesicular trafficking [18,83]. Vacuolar H^+^-ATPase, a proton pump localized on the lysosomal limiting membrane, enables H^+^ flux into the lysosomal lumen to maintain a pH of ~4.6 [84]. The luminal concentration of H^+^ to achieve this pH is 25 μM (Figure 2), and roughly 500 times higher than cytosolic H^+^ levels, which is necessary for lysosomal trafficking and content condensation during membrane fission [84]. Importantly, lysosomal luminal H^+^ content, luminal pH and activity of the vATPase proton pump are determined by the location of lysosomes and nutrient status of the cell [85]. Studies in Hela cells suggest that peripheral lysosomes are more alkaline than juxtanuclear lysosomes [85]. 

### 4.3. Na^+^/K^+^

Prior studies suggested that when compared to, Na^+^ content is lower in the lysosomal lumen compared to K^+^ ionic content [86]; however, using ultracentrifugation and mass spectroscopy, it was later deduced that luminal Na^+^ concentration is actually higher than K^+^ [87]. The luminal Na^+^ concentration ranges from 20 to 140 mM, whereas K^+^ concentration ranges from 2 to 50 mM (Figure 2), making Na^+^ the most abundant cation within the lysosome [87]. Lysosomal uptake of Na^+^ in exchange for a proton occurs via Na^+^/H^+^ exchangers (NHEs) such as NHE3, NHE5 and NHE6 [84,88]. The major function of the Na^+^ cation is to maintain the lysosomal membrane potential ∆ψ = ψ_cytosol_ − ψ_lysosome_, determined by the ionic gradient between the lysosomal lumen and cytosol, as well as ionic permeabilities across the lysosomal membrane [11]. Increased Na^+^ and K^+^ flux within the lysosome hyperpolarizes the lysosomal membrane, perturbing lysosomal luminal H^+^ gradient and acidification [11]. 

### 4.4. Cl^−^

The lysosomal luminal Cl^−^ concentration is estimated to be ~60–80 mM (Figure 2), the most abundant anion within the lysosome [89]. Among the Cl^−^ transporters, ClC-7 is the primary transporter localized on the lysosome and is ubiquitously expressed, whereas ClC-6 is a second major lysosomal Cl^−^ transporter that is expressed in the central and peripheral nervous system [89,90]. Silencing ClC-7 in Hela cells suppresses Cl^−^/H^+^ antiporter activity and impairs lysosomal luminal acidification suggesting that ClC-7 is important for lysosome function [84,90]. However, loss of ClC-7 in neurons has been implicated in the pathogenesis of lysosomal storage diseases and in neurodegeneration independent of its effect on lysosomal pH [86,91]. These studies indicate that ClC-7-mediated effect on lysosomal luminal pH is likely cell-specific.

### 4.5. Fe^2+^ and Zn^2+^

Fe^3+^, Fe^2+^, Zn^2+^ and Cu^2+^ are in micromolar concentrations within the lysosomal lumen (Figure 2). Metal ions can be released from the lysosome via proteolysis or via endocytosis or autophagocytosis of metal-bound proteins [92]. Fe^2+^ ion-dependent increases in ROS level aids in the survival of lysosome-resident pathogens; therefore, the levels of Fe^2+^ ions are under tight regulation through its binding to ferritin. [93]. Lysosome-autophagy is important in regulating intracellular levels of Fe^2+^. By the process of ferritinophagy, nuclear receptor coactivator 4 (NCOA4) recognizes and delivers ferritin bound Fe^2+^ to the lysosome [94,95,96,97]. Within the lysosome, ferritin is degraded and Fe^2+^ is released into the cytosol through the TRPML1 channel in both mammals and flies [96,98]. Similarly, intracellular Zn^2+^ levels are also regulated by Zn^2+^ transporters ZnT2 and ZnT4, which deliver Zn^2+^ to the lysosome [99,100]. Subsequently, Zn^2+^ is retrogradely transported into the cytosol through TRPML1 channel [101]. Furthermore, loss of function of TRPML1 leads to lysosomal Zn^2+^ deposition, triggering growth defects, impairing immune responses and causing neurogenerative diseases [101]. Therefore, lysosome activity-dependent regulation of cell proliferation requires iron homeostasis [94]. 

## 5. Role of Ion Channels in Lysosomal Function

Ionic movement is maintained by channels and transporters located on the lysosomal membrane, which is pertinent to ensure proper ionic homeostasis within the lysosomal lumen. There are three lysosomal Ca^2+^ channels in mammals: TRPML1-3, TPC1-2 and P2X4 which play a key role in regulating lysosomal activity and function (Figure 2).

### 5.1. Transient Receptor Potential Cation Channel, Mucolipin Subfamily (TRPML) 

Mammalian mucolipin family belonging to the large superfamily of transient receptor potential (TRP) consists of three non-selective cation channels, TRMPL1, TRPML2 and TRPML3. TRPML1 is exclusively localized to lysosomes and is ubiquitous, TRPML2 is localized on endosomes, and TRPML3 is found on both lysosomes and endosomes [102,103]. Structurally, TRPML channels consist of four subunits, which include cytosolic N- and C-termini with six-transmembrane domains. TRPML channels are permeable to Ca^2+^, Na^+^, K^+^, Fe^2+^ and Zn^2+^ (Figure 2) [11]. Upon activation, TRPML channel releases Ca^2+^ from the endo-lysosomal lumen to regulate various physiological processes such as endo-lysosomal membrane formation, phagocytosis, lysosome biogenesis, autophagy and exocytosis. Lysosomal Ca^2+^ efflux via mucolipin-1 (MCOLN1) activates calcineurin, leading to phosphatase-dependent increases in nuclear translocation of transcription factor EB (TFEB), that subsequently upregulates lysosome biogenesis and autophagy [104]. Additionally, increases in reactive oxygen species (ROS) can activate TRPML1 channels to facilitate lysosomal Ca^2+^ release and promote nuclear translocation of TFEB, which scavenge ROS via lysosomal autophagy [105]. TRPML channels have also been implicated in regulating lysosomal exocytosis, as described later in this review. 

Mucolipidosis type IV (MLIV) is a genetic lysosomal storage disorder (LSD) in children resulting in neurodegeneration and muscular dystrophy due to a loss of function mutation in TRPML1. Similarly, MLIV deficiency in Drosophila inhibits autophagy with concomitant accumulation of apoptotic bodies, whereas TRPML1 overexpression rescues this phenotype [106]. Furthermore, TRPML1 and PI(3,5)P2 (phosphatidylinositol 3,5-bisphosphate; an activator of TRPML1) signaling is impaired in Nieman-Pick disease (NP) and many other LSDs [107]. Overexpression or increased activity of TRPML1 augments lysosomal Ca^2+^ release, reverses LEL-Golgi trafficking defects and reduces cholesterol accumulation in NPC cells [108]. Unlike TRPML1, TRPML2 regulates recycling of glycosylphosphatidylinositol-anchored proteins (GPI-APS) through its effect on the small GTPase Arf6 [109], which is implicated in regulating the innate immune response [110]. Mice deficient in both TRPML1 and 3 exhibit growth delay, a phenotype that is not present when only one TRPML isoform is deficient [111]. Collectively, these findings suggest that TRP channels regulate lysosome-dependent processes with therapeutic potential to treat LSDs and lipid storage disease. 

### 5.2. Two-Pore Channels (TPC)

Mammals and rodents have two isoforms of TPCs, TPC1 and TPC2, whereas lower vertebrates have the additional isoform TPC3. TPCs span a dual six-transmembrane domain, which exhibits sequence homology with voltage-gated Na^+^ and Ca^2+^ channels with selective impermeability to K^+^ ions and permeability to Ca^2+^, Na^+^ and H^+^ ions (Figure 2) [112,113]. TPC channels are important for vesicular endocytosis and autophagy function [114,115]. TPCs also regulate NAADP (nicotinic acid adenine dinucleotide phosphate)-dependent mobilization of lysosomal luminal Ca^2+^ stores [116,117]. NAADP-dependent lysosomal Ca^2+^ release is abrogated in MEFs and macrophages isolated from TPC1 or TPC2 knockout mice [117]. Similarly, pancreatic β-cells isolated from TPC2-knockout mice fail to activate NAADP-dependent cation current [116]. Hearts from TPC2-knockout mice exposed to isoproterenol for 14 days are less hypertrophic, less vulnerable to develop ventricular arrythmias and show improved cardiac function, an observation which also requires functional CaMKII. Additionally, TPC2/NAADP-dependent lysosomal Ca^2+^ release increases the cytosolic level of Ca^2+^ released from the ER/SR, suggesting that TPC2/NAADP contributes to the action of β-adrenoceptor signaling likely by triggering Ca^2+^-induced Ca^2+^ release in the heart [118]. These findings support the notion that TPCs play an important role in NAADP-dependent Ca^2+^ signaling and indicate that TPCs bind to NAADP with high affinity. 

### 5.3. P2X4

P2X4 receptor belongs to the family of purinoceptors and is a Ca^2+^-permeable channel (Figure 2) widely expressed in the central and peripheral nervous system, epithelial cells and smooth muscle cells [119]. P2X4 is primarily localized within the lysosomal compartment, wherein they are targeted via di-leucine and tyrosine motifs and trafficked to the plasma membrane upon lysosome exocytosis [120]. Functionally, P2X4 plays an important role in promoting endo-lysosomal membrane fusion in a Ca^2+^/calmodulin-dependent manner [121]. Unlike TRPML and TPC, P2X4 channel is activated during alkalization of the lysosomal lumen but remains inactive in the acidic lysosomal milieu [122,123]. 

## 6. Contact Site Between the Lysosome and other Organelles

Lysosomes develop membrane contact sites with other organelles to transfer signaling information, to share metabolites and to facilitate ionic homeostasis. This inter-organelle communication, coupled with metabolic changes, profoundly impacts lysosomal function and cellular homeostasis. 

### 6.1. Lysosome-ER Contact Site 

Membrane contact sites (MCS) are zones of close apposition (~30 nM) within cells that are used for inter-organelle communication [124]. Lysosome-ER MCS is implicated in Ca^2+^ mobilization and signaling, endosome trafficking, maturation, lysosomal biogenesis [125] and positioning [123,126]. Using high-resolution three-dimensional electron microscopy, Friedman et al., demonstrated that ER tubules enclose around endosomes and mobilize, as they remain in contact with microtubules, to facilitate EEs maturation to endo-lysosomes [125]. Notably, 50% of EEs and 95% of LEs form a contact site with the ER [125]. Lysosome-ER MCS mobilizes cholesterol (FC) generated within the lysosome via hydrolysis of LDL cholesterol ester to the ER. The concerted action of lysosomal membrane protein (NPC1), ER membrane proteins (VAPA (vesicle-associated membrane protein (VAMP)-associated ER protein A) and VAPB) and lipid-transfer proteins (ORP1L (oxysterol-binding protein-related protein 1L) and ORP5) is necessary for mobilizing FC from the lysosome to the ER (Figure 3A, step 1). ORP5 silencing leads to cholesterol accumulation within the lysosomal lumen, disrupting lysosomal acidification and function [127,128].

The ER and lysosome contact site is also implicated in regulating lysosomal positioning [126,129]. Low cellular cholesterol promotes lysosome-ER MCS formation activating ORP1L, which triggers the anchoring of VAPA with Rab7-RILP (Rab7-interacting protein) (Figure 3A, step 2). Subsequently, perturbation of lysosomal dynein-dynactin interaction induces LEs accumulation within the cell periphery towards the plus end of the microtubule [126]. In contrast, increased cellular cholesterol inhibits ORP1L-VAP interaction and formation of lysosome-ER MSC, enabling dynein-dynactin-dependent mobilization of LEs towards the center of the cell i.e. at the microtubule minus end [126]. In Niemann-Pick disease type C, cholesterol accumulation decreases the movement of lysosomes in both directions, leading to lysosomes aggregating at the perinuclear region [130]. 

Another important function of the lysosome-ER MCS is the regulation of Ca^2+^ flux between the two organelles. Lysosomal Ca^2+^ release mediated by its secondary messenger nicotinic acid adenine dinucleotide phosphate (NAADP), acts as an IP3 receptor agonist that in turn triggers Ca^2+^ release from the ER (Figure 3A, step 3) [131,132,133,134,135]. The regulation of ER Ca^2+^ release by the lysosome has been demonstrated in multiple studies [124,136]. Glycyl-L-phenylalanine-2-naphthylamide (GPN) a lysosomotropic agent, evokes lysosomal Ca^2+^ efflux, which is sufficient to induce a secondary spike in Ca^2+^ release from the ER into the cytosol in primary-cultured human skin fibroblasts [124,136]. Interestingly, Ca^2+^ release from the ER can also influence the efflux of lysosomal Ca^2+^. For instance, in sea urchin egg, ER Ca^2+^ release recruits NAADP to the lysosome, activating TPC channels to induce Ca^2+^ efflux from the lysosome [137]. These findings highlight a bidirectional regulation of Ca^2+^ signaling occurring between the lysosome and the ER. Proximity ligation assay experiments further indicate a close association between IP3R and LAMP1/Rab7 (marker for late endosome/lysosome), suggesting that bidirectional Ca^2+^ signaling also requires the formation of lysosome-ER MCS [124,137]. ER-lysosome proximity can be disrupted by inhibiting lysosome acidification. By treating HEK293 cells with v-ATPase inhibitor, interaction between IP3R and LAMP1/Rab7 is reduced, causing the lysosome to redistribute and enlarge, perturbing ER-Ca^2+^ exchange [17]. In contrast, deletion of IP3R isoforms does not affect lysosome-ER proximity, indicating that lysosomal luminal pH is crucial for the formation of lysosome-ER MCS and for bidirectional regulation of Ca^2+^ signaling between organelles [17]. 

### 6.2. Lysosome-Mitochondria Contact Site

Mitochondrial and lysosomal membranes are apart from each other by a distance of ~10 nm, making it ideal for MCS formation between the two organelles [138,139]. Approximately 15% of lysosomes form a contact site with mitochondria [138]. Recently, Wong et al., demonstrated that lysosomes and mitochondria form a stable MCS, which is induced by the lysosomal GTP-bound Rab7 protein, highlighting the bidirectional regulation of lysosome and mitochondrial dynamics [139]. Alternately, uncoupling of lysosome-mitochondria MCS is promoted by hydrolysis of Rab7-GTP driven by the Rab7-GAP protein, TBC1D15 (TBC1 Domain Family Member 15), which is recruited by the mitochondrial protein Fis1 (mitochondrial fission 1 protein) [139]. However, in HeLa cells, mutation in Rab7, Fis1 or TBC1D15 prevents the recruitment of TBC1D15 to the mitochondria, resulting in an increase in the duration of mitochondria-lysosome contact site, causing accumulation of enlarged lysosomes and impairment in mitochondrial fission events (Figure 3B, step 1) [139]. However, the lysosome-mitochondria MCS formation was not prevented by the penta knockout (p62, NDP52, OPTN, NBR1 and TAX1BP1) of mitophagy markers in Hela cells, suggesting that MCS formation is independent of the process of mitophagy [138]. 

Both lysosome function and lysosomal Ca^2+^ are critical to regulate mitochondrial function and homeostasis [16,105,139]. The TRPML1 channel located on the lysosome, releases Ca^2+^ in response to increases in mitochondrial reactive oxygen species (ROS). To scavenge ROS and damaged mitochondria, lysosome biogenesis and autophagy is increased by ROS-dependent activation of calcineurin and ensuing nuclear translocation of TFEB (Figure 3B, step 2) [105]. Prior study demonstrates that deficiency in mitochondrial function in CD4^+^ T lymphocytes by genetic deletion of mitochondrial transcription factor A (Tfam), results in impaired lysosome function and autophagy [140]. Using both an in vitro model and an in vivo mouse model of respiratory chain dysfunction, it was shown that acute mitochondrial stress increases lysosome biogenesis via an AMPK-TFEB/MITF-dependent pathway [141]. However, it is unclear whether the mutual relationship between lysosome and mitochondria to regulate biogenesis, metabolism and function requires the formation of lysosome-mitochondria membrane contact site. 

## 7. Lysosome Fusion and Fission Events 

### 7.1. Late Endosome-Lysosome-Autophagosome Fusion 

Macromolecules are transported via secretory vesicles, or an endocytic, autophagic or phagocytic pathway for degradation. Since LEs and lysosomes are located near the microtubule-organizing center, the bulk of lysosome-endosome fusion occurs in the juxtanuclear region. Organelle fusion ensues via tethering processes when two organelles form a contact site over a distance of ~25 nm [142]. Tethering of lysosomes and endosomes requires the small GTPase Rab7, which interacts with RILP (Rab7-interacting protein) and recruits HOPS (mammalian homotypic fusion and vacuole protein sorting ) complex (Figure 3C, step 1) [143,144]. Notably, Rab7, coupled with VPS18 and VPS39, also promotes tethering of LEs and lysosomes (Figure 3C, step 2). Alternatively, overexpression or silencing of either Rab7, VPS18 or VPS39 in HeLa cells results in organelle clustering and organelle dispersion, respectively [145,146,147]. In addition to the small GTPase Rab7, formation of the HOPS complex assembly on a lysosome requires Arf-like8b (Arl8b) [148]. Indeed, co-binding of Rab7 and Arl8b to Rab7 effector PLEKHM1 (pleckstrin homology domain-containing protein family member 1) induces clustering and heterotypic fusion of LEs-lysosomes (Figure 3C, step 3) [144,148,149]. Fusion of lysosomes-LEs also requires Trans-SNARE (soluble N-ethylmaleimide-sensitive factor attachment protein receptor) assembly, composed of three clustered Q-SNARE (Qa, Qb and Qc) elements, which interact with R-SNARE through their N-terminal end of the SNARE motif [150]. Notably, synaptotagmin I and complexin proteins exert control over the Trans-SNARE complex. Synaptotagmin I triggers Ca^2+^-dependent fusion of organelles by using its two Ca^2+^-binding C2 domains, while complexin binds to the surface of the SNARE complex in a Ca^2+^-dependent manner to regulate organelle fusion [151]. Furthermore, antibody-based function-blocking experiments provided evidence that Q-SNARE such as syntaxin-7 (Qa-SNARE), VTI1B (VPS10 tail interactor-1B, Qb-SNARE) and syntaxin-8 (Qc-SNARE) are required for endosome-lysosome heterotypic fusion. R-SNARE requires VAMP7 (vesicle-associated membrane proteins) and VAMP8 for both heterotypic and homotypic (fusion of lysosomes) events (Figure 3C, step 4) [82,152]. 

Lysosomes also fuse with autophagosomes, which is critical for macromolecule degradation via formation of autolysosomes [153]. A well-coordinated transport of lysosomes and autophagosomes to the perinuclear region of the cell is required for effective fusion between two organelles [154,155,156]. During starvation, an increase in intracellular pH causes mobilization of autophagosomes and lysosomes via microtubules to the perinuclear area [85,157]. Both HOPS complex and Rab7 effector protein PLEKHM1 integrate endocytic and autophagic pathways at the lysosomal membrane [158,159]. At least in HEK293 cells, the interaction with autophagosomal membranes is facilitated by PLEKHM1 binding to HOPS via the LC3-interacting region (LIR) (Figure 3D, step 1) [159]. Silencing PLEKHM1 in Hela and HEK293 cells failed to activate autophagy upon mTOR inhibition, suggesting that PLEKHM1 is crucial for autophagosome-lysosome fusion [158,159]. Similar to PLEKHM1, BLOC-1 related complex (BORC1) is implicated in autophagosome-lysosome fusion [160]. In non-neuronal cells, BORC1 facilitates recruitment of Rab7-HOPS (Figure 3D, step 2) for tethering and enables Trans-SNARE complex formation of STX17-VAMP8-SNAP29 to induce autophagosome-lysosome fusion (Figure 3D, step 3), whereas knockdown of BORC disrupts autophagosome-lysosome fusion [160]. Interestingly, Nguyen et al., demonstrated that autophagosome formation and selective sequestration of mitochondria in HeLa cells is regulated by GABARAP but not STX17 [161]. Additionally, cytosolic Ca^2+^ augments lysosomal fusion to the plasma membrane to trigger lysosome exocytosis, which is described later in the review. 

### 7.2. Lysosomal Exocytosis (Fusion of Lysosomes with the Plasma Membrane) 

Lysosomes also regulate secretory vesicles through lysosome exocytosis. Exocytosis of the lysosome involves two main steps: (1) lysosomal movement to the cell periphery for plasma membrane (PM) fusion, and (2) exocytosis of the luminal content of the lysosome into the extracellular milieu. Lysosomal exocytosis is paramount for various cellular physiological processes such as PM repair (PMR), immune response, bone resorption and cell signaling [162]. PM integrity is lost upon injury, resulting in unlimited exchange of intra- and extra-cellular components triggering cell death; thus, PM repair is crucial for the maintenance of cellular homeostasis. Upon PM injury, cytoskeletal and motor proteins induce trafficking of lysosomes to the damaged PM at the cell periphery, wherein lysosomes fuse directly with the PM and empty their content into the extracellular space [107]. PM damage induced by exposing Hela cells to mechanical stress, triggers the colocalization of the motor protein KIF5B (Kinesin5 subfamily member) with LAMP1-positive lysosomes, which are then mobilized towards the cell periphery (Figure 4, step 1) [163]. Depletion of KIF5B in Hela cells causes lysosome immobilization and peripheral lysosome aggregation, signifying that motor proteins possess an important role in lysosomal mobilization during exocytosis [163]. 

Amongst the different ways cells adapt to various effector stimuli is a cell’s capacity to sense increases in intracellular Ca^2+^ level and respond by triggering lysosomal exocytosis (Figure 4, step 2) [164]. The rate of increase in Ca^2+^ influx following PM damage is sensed by synaptotagmin-VII (Syt-VII) at the lysosomal membrane (Figure 4, step 3) [164]. Subsequently, Syt-VII interacts with Trans-SNARE complex on the PM, which is formed by the interaction of lysosome-localized VAMP7 (Vesicle-associated membrane protein 7) with syntaxin-4 and SNAP23 (synaptosome-associated protein 23 kDa). To facilitate fusion, trans-SNARE complex brings lysosomes into close proximity with the PM (Figure 4, step 4) [165]. Indeed, treatment with ionomycin suppresses lysosome exocytosis in embryonic fibroblasts isolated from Syt-VII KO mice expressing negative-dominant VAMP7 construct, [165], suggesting that Trans-SNARE complex and Syt-VII exert important functions in membrane fusion and exocytosis. Following membrane fusion, exocytosis causes efflux of the lysosomal enzyme sphingomyelinase (aSMase), which converts sphingomyelin into ceramide on the PM (Figure 4, step 5) [166]. This results in an inward curvature of the PM to facilitate endocytosis-mediated removal and resealing of the damaged PM (Figure 4, step 6) [166,167], an effect impaired in cells deficient in aSMase. Although lysosomal exocytosis is permissible, the ability to facilitate PM repair through endocytosis is suppressed in aSMase deficient cells [166,167], suggesting that aSMase is critical for lesion internalization and repair. 

TRPML1 (mucolipin-1), a Ca^2+^-permeable non-selective cation channel is a driver of lysosome exocytosis [164,168]. MCOLN1 gene (encodes mucolipin-1, MLN1) mutations cause Mucolipidosis type IV (MLIV) characterized by late endosome/lysosome accumulation. Dermal fibroblasts from MLIV patients display decreased TRPML1 channel activity and impaired lysosome exocytosis as measured by the rate of NAG (*N*-acetyl-β-d-glucosaminidase) enzyme release following ionomycin treatment, effects which are rescued by TRPML1 overexpression [168]. Collectively, these findings suggest that MLN1 possesses an important role in regulating lysosome exocytosis. 

### 7.3. Lysosome Fission

Lysosome fission is crucial for maintaining steady-state levels of lysosome number and size [169,170]. Lysosome fission involves vesiculation, tubulation and “kiss-and-run” events [13]. Proteins involved in the vesiculation process include an outer-coat protein clathrin, an inner-coat adaptor protein known as AP, and the motor protein dynamin 1 [169]. Initiation of vesiculation occurs via cargo sorting, a process that requires binding of adaptor protein (AP) complex to the lysosomal membrane. Subsequently, clathrin and its structural protein scaffolds, sec 13/31 COPII subcomplex, curve and deform the membrane [169]. These steps culminate in activation of scission machinery proteins such as dynamin 1, a GTPase or AAA-ATPase Vps4 of ESCRT-III complex or BAR (brefeldin A ADP-ribosylated substrate) domain protein scaffold, catalyzing the membrane fission by dissociating vesicles from the lysosomal membrane (Figure 5A) [169,170]. 

After vesiculation, the process of tubulation recycles and reforms lysosomes via tubular extrusion. Tubulation is initiated by the coat protein clathrin, which deforms and curves the membrane to form the tip of the nascent tubule. During periods of prolonged starvation, initial decline of cellular lysosome content occurs concomitantly with autolysosome (AL) formation, followed by de novo replenishment of the lysosomal pool. This process of de novo formation of lysosomes is termed as autophagic lysosomal reformation (ALR) [171]. ALR theory best describes the lysosome tubulation process. Rat kidney cells subjected to 4 h of nutrient deprivation stained positive for LAMP1 (marker for lysosome) and LC3 (marker for AL), whereas cells starved beyond 12 h stained positive only for LAMP1 and not LC3 [172]. Therefore, acute starvation augments AL content, whereas chronic starvation restores lysosome content to levels observed before starvation [172]. Furthermore, “proto-lysosomes”, which are non-acidified and empty extended tubular structures, were observed on LAMP1-YFP positive ALs by 8 hof starvation [172]. However, by 12 hof starvation, proto-lysosomes mature into functional lysosomes by acquiring acidity and degradative capacity via the process of ALR (Figure 5B) [172]. 

Co-localization studies and clathrin-knockdown approaches demonstrated that clathrin recruitment is increased 12 h post starvation to regulate ALR [173]. Proteomic analysis and SAMCell-RNAi screening identified clathrin-heavy chain and phosphatidylinositol-4-phosphate 5-kinase (PIP5K1B) as candidate protein regulating ALR [173]. Furthermore, PIP5K1B converts PI4P (phosphatidylinositol-4-phosphate) into PI4,5P2 (phosphatidylinositol-4,5-biphosphate) on ALs to trigger clathrin recruitment via adaptor protein AP2 [172,173,174]. During acute starvation, PIP5K1B silencing leads to enlargement in LC3-CFP- and LAMP1-YFP-positive ALs impairing clathrin-dependent tubule reformation and ALR [172,173]. Furthermore, during prolonged starvation, amino acid transport from ALs to the cytoplasm restores mTOR activity and inhibits autophagy with concomitant generation of LAMP-1-positive proto-lysosomal tubules extruding from the ALs [172]. Interestingly, impaired mTOR reactivation and loss of ALR is observed in fibroblasts from patients with Pompe and Niemann–Pick disease [172]. These data indicate that mTOR couples cellular nutrient status to autophagic functioning through its regulation on ALR and lysosome numbers. 

Lysosomal fission events culminate in the “kiss-and-run” fusion process. Initially, fusion of autophagosomes and late endosome/lysosome occurs to generate a fully matured, hybrid organelle. However, multiple organelles fuse frequently to exchange their content despite being separate vesicles, denoted as a “kiss-and-run” event [175]. Indeed, Bright et al., elegantly demonstrated the kiss-and-run content mixing model in a living cell by labeling endosomes and lysosomes with Oregon green 488 and rhodamine dye, respectively [175]. A transient interaction occurs between green-endosome and red-lysosome with progressively augmented red-over-green fluorescence intensity, indicating a “kissing” event that eventually terminates in separation of a red-lysosome, highlighting a kiss-and-run delivery of red fluorescence independent of direct fusion (Figure 5C) [175]. 

## 8. Lysosome Localization and Movement

Intracytoplasmic movement of the lysosome is critical for executing processes such as lysosome biogenesis, protein quality control and metabolism to ensure cellular homeostasis. Altered lysosomal acidification [157,176], misfolded protein aggregate formation [177] and cellular nutrient status [178] perturb lysosome positioning and motility. During fed conditions, lysosomes are peripherally localized, whereas after nutrient depletion, lysosomes migrate towards peri-nuclear regions to facilitate autophagosome-lysosome fusion. The mobilization of lysosomes from the cell periphery towards the perinuclear region is termed “retrograde movement”, while movement of lysosomes from the perinuclear area towards the cell periphery is termed as “anterograde movement” [179,180,181]. Lysosomal movement in the cell is assisted by cytoplasmic dynein motor proteins and activator protein dynactin, and with kinesin motor proteins [179,180,181]. Lysosomes exhibit bidirectional movement along microtubular platforms, which are regulated by microtubule motor proteins. Dynein motor proteins, with associated dynactin proteins, allow retrograde movement of lysosomes from the plus end to the minus end of microtubules [143,182]. Rab7 GTPase and its downstream target RILP are obligatory for dynein-dynactin recruitment to the lysosome and for transport of lysosomes to the central region of the cell (Figure 6, step 1) [143]. Moreover, TBC1D15 (GTPase-activating protein, GAPs) and TBC1D2 [181,183] govern Rab7-dependent recruitment of dynein-dynactin on the lysosome. Kinesin superfamily (KIF) proteins, which includes kinesin-1 (KIF5A, KIF5B and KIF5C) [184], kinesin-2 (KIF3) [185] and kinesin-3 (KIF1A, KIF1B) [186], trigger the anterograde movement of the lysosome from the minus end to the plus end along the microtubule and ultimately regulate lysosome positioning (Figure 6, step 2) [179,180,181]. Similarly, BLOC-1-related complex (BORC), a multicomplex protein consisting of eight subunits, is indispensable for lysosome positioning and movement. BLOC-1 interacts with the lysosomal membrane through Arf-like small GTPase Arl8 (Figure 6, step 3) [187]. Subsequently, a signaling cascade is initiated via Kinesin-interacting protein (SKIP) to promote lysosome migration towards the cell periphery [187]. 

Intraluminal acidification of the lysosome is dependent on the intracellular localization and movement of the lysosome [85,176]. Quantitative ratiometric fluorescence microscopy reveals that peripheral lysosomes are more alkaline compared to centrally localized lysosomes. Peripheral lysosomes exhibit higher proton leakage and reduced vATPase activity compared to the centrally-localized lysosomes [85]. Furthermore, peripheral lysosomes display increased Arl8b expression and recruitment of kinesin-1 on their membrane, whereas Rab7 abundance and recruitment of RILP are reduced [85]. On the contrary, lysosomes at the perinuclear region exhibit higher vATPase activity and increase in Rab7-dependent recruitment of RIPL on lysosomes [85]. Due to higher vATPase activity, perinuclear lysosomes are preferentially used for proteolytic degradation [85]. Together, these findings suggest that lysosome localization influences vATPase activity, which is crucial for maintaining intraluminal pH.

Cellular nutrient status also influences lysosome localization and motility to regulate mTORC1 activity. During nutrient-rich conditions, mTORC1 is activated on peripheral lysosomes to ensure close proximity to signaling receptors at the cell surface [178]. Alternatively, starvation represses mTORC1 activity and induces perinuclear clustering of lysosomes due to changes in luminal pH of the lysosome, which in turn is critical for autophagosome formation and autophagosome-lysosome fusion [178]. Starvation induces transcription of genes encoding for TRPML1 and TMEM55B by activating TFEB and TFE3. Retrograde mobilization of the lysosome is induced in a dynein-dynactin manner by TRPML1 and TMEM55B, which interact with C-Jun-amino-terminal kinase-interacting protein 4 (JIP4) (Figure 6, step 4) [188]. A recent study demonstrated that TRPML1 activation induces biogenesis of autophagic vesicles by generating phosphatidylinositol 3-phosphate (PI3P) and recruiting PI3P-binding proteins to the nascent phagophore [189]. TRPML1 activity is also required for Ca^2+^-dependent centripetal transport of the lysosome towards the perinuclear region in HeLa cells and Cos1 cells, an effect independent of the Rab7-RIPL pathway [78]. Furthermore, TRPML1 activity requires the interaction between ALG-2 (an EF-hand-containing protein, serve as lysosomal Ca^2+^ sensor) and dynein-dynactin for retrograde mobilization of the lysosome (Figure 6, step 5) [78]. Notably, increases in phagophore formation by TRPML1 requires signaling by calcium-dependent kinase CaMKKβ and AMPK, which activates ULK1 and VPS34 autophagic protein complexes[189]. Indeed, mouse fibroblasts treated with Ca^2+^ chelator BAPTA-AM, demonstrate significantly reduced retrograde and anterograde lysosomal motility, suggesting that Ca^2+^ is important for regulating lysosome positioning and movement [78]. 

## 9. Lysosome-Dependent Protein Degradation

Lysosomes contain luminal proteases, lipases, glycosidases and nucleases that degrade intracellular and extracellular macromolecules via autophagy [190]. To maintain cellular energy balance and organelle homeostasis, lysosomal autophagy (self-eating/digestion) degrades and recycles both short- and long-lived proteins, protein aggregates, mitochondria (mitophagy) [191], nuclei (nucleophagy) [71], endoplasmic reticulum (reticulopathy) [192], lipid droplets (lipophagy) [190], and peroxisomes (pexophagy) [193]. Two primary types of autophagy are described in mammalian cells, macroautophagy and chaperone-mediated autophagy (CMA). A less characterized type of autophagy is microautophagy (MIA) involving pinocytosis of cytosolic regions surrounding lysosomes thereby engulfing and degrading cytosolic cargo. 

### 9.1. Macroautophagy 

Macroautophagy occurs within an ER-derived double membrane-bound organelle, known as an autophagosome, which allows the bulk delivery of proteins and organelle cargo to lysosomes for proteolysis. Autophagy related gene (Atg) proteins organize and control various steps of macroautophagy [194,195]. Macroautophagy is initiated through the formation of an ER-derived pre-autophagosome, containing partially selected bulk sequestration of protein cargo and organelles. Pre-autophagosome assembly requires the functional class III phosphatidylinositol-3-kinase (PI3K) complex [196,197]. Elongation and maturation of the pre-autophagosome requires the ubiquitin-like conjugation system. Autophagosome closure and maturation is governed by the lipidation of microtubule-associated protein 1 light chain 3 (LC3I) to form LC3II. Subsequently, autophagosomes fuse with lysosomes forming autolysosomes, wherein lysosomal proteases degrade intra-autophagosomal content [198]. In this section, we have only superficially addressed different types of autophagic processes as they are comprehensively reviewed elsewhere [195,199]. 

### 9.2. Lipophagy, Mitophagy and Lysophagy 

Lysosomal autophagy also plays an important role in breaking down lipid droplets via a process referred to as “lipophagy” [200]. Lipophagy was first described in hepatocytes and is indispensable for TAG breakdown as Atg7^−/−^ mice displayed hepatic TG accumulation [201,202]. In addition to lipophagy, mitophagy ensures mitochondrial quality control by degrading dysfunctional mitochondria and maintaining functionally-active mitochondria [203,204,205]. Dysfunctional mitochondria that are unable to handle ROS overload are targeted for mitophagy by a PTEN-induced putative kinase 1 (PINK1)-Parkin mediated mechanism [206]. Specifically, PINK1 protein content is increased on the mitochondrial outer membrane, which signals Parkin-mediated ubiquitination of PINK1 molecule [206]. Ubiquitinated PINK1 molecules are targeted by ubiquitin cargo receptors p62/SQSTM1 and NBR1 [207,208]. Similar to mitochondria, damaged lysosomes are also subjected to repair or clearance albeit with assistance from ESCRT machinery-dependent sorting complex. The process of degrading damaged lysosomes via selective macroautophagy is termed as “lysophagy” [209,210,211]. Impaired lysosomal homeostasis is observed when lysosomes are damaged due to increased lysosomal membrane permeabilization, or lysosomal stress induced by lysosomotrophic agents [212,213]. Upon injury, damaged lysosomes efflux their luminal Ca^2+^ stores and activate the lipid-binding activity of the ESCRT complex, which is essential to repair lysosomes by reacidifying the lysosomal lumen and replenishing the lysosomal hydrolases [210]. Prior studies suggest that damaged lysosomes expose luminal glycan to the cytosol, which is sensed by galectin-3 and recruits E3 ubiquitin ligase TRIM16 to trigger ubiquitination of the damaged lysosomes [209,214]. Following the ubiquitination of lysosomes, the autophagic machinery is activated to initiate lysophagy (Figure 7) [209,214]. 

### 9.3. Chaperone-Mediated Autophagy (CMA) 

CMA is initiated ~10 h after starvation and, unlike macroautophagy, promotes selective protein degradation [215,216]. CMA targets ~35% of soluble cytosolic proteins, which harbor a KFERQ (composed of sequence of amino acids includes lysine (K), phenylalanine (F), glutamic acid (E), arginine (R) and glutamine (Q)) consensus motif, for lysosomal degradation [215]. Cytosolic chaperones, heat-shock cognate protein of 70 kDa (hsc70) and heat shock protein 90 (Hsp90) recognize the KFERQ motif and deliver targeted proteins to the lysosomal surface for binding to lysosome-associated membrane protein-2A (LAMP-2A) [217,218]. Upon binding to LAMP-2A, the substrate protein is unfolded, translocated and rapidly degraded within the lysosomal lumen. Given that the proteins undergoing degradation through CMA exhibit diverse roles in a myriad of intracellular processes, it is plausible that changes in LAMP-2A function and lysosomal biogenesis could have a significant impact on degradation of CMA-targeted proteins and lead to significant impacts on cellular metabolism and function.

### 9.4. Microautophagy

Microautophagy is a process wherein direct invagination of lysosomes facilitates intracellular degradation of protein and organelle cargo. New studies have identified the role of microautophagy in processing cargos through peroxisomes (micropexophagy) [219], mitochondria (micromitophagy) [220] and lipid droplets (microlipophagy) [221]. 

## 10. Lysosome Dysfunction in Diseases

Lysosomal storage disorders (LSDs) are a group of disorders caused by inherited mutations of genes involved in lysosomal function, leading to macromolecule accumulation within lysosomes. Additionally, defects in lysosome function have been identified in other conditions including cancer, cardiovascular disease and metabolic disorders. 

### 10.1. Lysosomal Storage Disorder (LSDs) 

LSDs are a family of 70 rare monogenic diseases that are clinically exist during infancy or childhood. LSDs affect 1 in 7700 live births [222]. Most LSDs represent mutations in the genes encoding lysosomal hydrolases, lysosomal membrane proteins, proteins involved in trafficking, and non-lysosomal proteins. LSDs are often associated with early-onset neurodegeneration and neurological sequelae are often triggered by undigested lysosomal macromolecules [4]. Depending on the accumulation of substrates, LSDs exhibit distinct phenotypes. For instance, a mutation in lysosomal enzyme β-glucocerebrosidase (GBA gene) causes Gaucher disease, characterized by organomegaly and cytopenia [223]. Deficiency in β-galactosidase (encoded by HEXA) and β-hexosaminidase (encoded by HEXB) result in the accumulation of gangliosides, characterized by severe neurological defects. Similarly, loss of function of HEX subunit A or B causes Tay-Sachs or Sandhoff disease [224]. Pompe disease or glycogen storage disease type II is characterized by lysosomal glycogen accumulation, caused by a deficiency of α-glucosidase enzyme (encoded by GAA gene) [225] resulting in cardiac and respiratory failure [226,227]. Similarly, defects in lysosomal membrane proteins such as NPC1, LAMP2 and MCOLN1 cause Niemann-Pick type C (NPC), Danon and Mucolipidosis type IV (MLIV) diseases [228,229], respectively. NPC disease is caused due to a mutation in either NPC1 or NPC2 gene, resulting in the intralysosomal accumulation of cholesterol, sphingolipid, glycosphingolipid, sphingosine and sphingomyelin [79,230]. Because cholesterol is the dominant form of stored lipid, these disorders are also termed “cholesterol storage diseases” [230]. Importantly, both NPC and MLIV are associated with impaired lysosomal Ca^2+^ homeostasis [228,229], given the decline in NAADP-mediated lysosomal Ca^2+^ release in NPC1 mutant human B lymphocytes and human fibroblasts [231]. Similarly, treating NPC1 mutant cells with myriocin reduces sphingosine deposition, restores lysosomal Ca^2+^ stores and reverses the NPC1 phenotype in WT cells [231]. Since two pore channels (TPC) have been implicated in regulating NAADP-dependent Ca^2+^ release [116], it is likely that sphingosine accumulation directly targets TPCs to alter lysosomal Ca^2+^ signal in NPC1 mutant cells. Defective Ca^2+^ homeostasis is also observed in Gaucher disease [232]. Microsomes isolated from the brain of a patient suffering from type II Gaucher disease exhibit 10 times higher amounts of glucosylceramide relative to control [233]. Ryanodine receptor activation increases Ca^2+^ release from the ER of neurons with stored glucosylceramide [234]. Similarly, GM2 accumulation in Sandhoff disease decreases the Vmax of sarco/endoplasmic reticulum Ca^2+^-ATPase pump (SERCA), altering the Ca^2+^ reuptake rate into the ER [235]. However, an outstanding question in the field is how intralysosomal lipid accumulation causes defects in ER morphology and Ca^2+^ homeostasis. It is possible that the lipid accumulation in the lysosome disrupts Ca^2+^ flux between the lysosome and ER, which likely trigger impairment in Ca^2+^ homeostasis. 

### 10.2. Neurodegenerative Diseases

Defective lysosomal function is also observed in neurodegenerative disorders such as Parkinson’s disease (PD), Alzheimer’s disease (AD), Huntington disease (HD), dementia with Lewy bodies, amyotrophic lateral sclerosis and Charcot-Marie-Tooth (CMT) disease. Lysosomes in neurodegenerative disorders exhibit disrupted and insufficient proteolysis [236], defective autophagosome-lysosome fusion [236], perturbed lysosome positioning [168] and altered lipid composition of the lysosomal membrane [237]. Failure in autophagosome clearance due to the defective lysosomal proteolytic activity and acidification is observed in AD with mutation in presenilin-1 (PS-1) [238]. Given that PS-1 is involved in the maturation of vATPase via its binding to V0a subunit, mutation in PS-1 impairs lysosomal autophagy and contributes to the pathogenesis of AD [238]. Deposition and aggregation of α-synuclein within Lewy bodies in PD and dementia is a result of dysfunctional lysosomal autophagy [236,239]. The conditional deletion of Atg7 in dopaminergic neurons induces accumulation of low-molecular-weight-α-synuclein and LRRK2 (leucine-rich repeat kinase 2), two proteins involved in the pathogenesis of idiopathic PD [240,241]. Conversely, activation of the lysosome-autophagy pathway by overexpression of TFEB or Beclin-1, or by pharmacological autophagy activator rapalog, attenuates neuronal deposition of α-synuclein-mediated toxicity [242]. In addition to macroautophagy, wild type α-synuclein is selectively targeted and translocated to the lysosomal lumen for degradation via CMA [243,244,245]. However, neuronal cells with mutations in α-synuclein, such as A53T and A30P, binds to CMA receptor LAMP-2A, preventing their own degradation as well preventing breakdown of other CMA substrates [243,244]. Overexpression of LAMP2A upregulates CMA and increases degradation of α-synuclein protein attenuating neurotoxicity [245]. 

Huntington disease (HD) is a genetic neurodegenerative disease caused by a mutation in Huntingtin (Htt) protein and displays characteristic defects in the lysosomal-autophagy pathway [246,247,248]. The polyglutamine-expanded huntingtin (polyQ-htt) protein is primarily degraded via lysosome-autophagy pathway [246]. Htt protein functions as a scaffold protein for selective macroautophagy. Htt interacts with cargo receptor p62, allowing it to bind to LC3 on autophagosomes [248]. Furthermore, both Htt and huntingtin-associated protein-1 (HAP1) colocalizes with autophagosomes and participates in maintaining autophagosome and lysosome dynamics in neurons [249]. By contrast, mutations in Htt compromise the ability of autophagic vacuoles to sequester cytosolic components, inducing cytotoxicity in HD [247,250]. Accumulation of lipid droplets and depolarized mitochondria in the cytosol is typically observed in neuronal and non-neuronal (MEFs) cells from HD subjects, highlighting the inability of lysosomal autophagy to recognize and degrade cytosolic content [250]. Defective lysosomal dynamics is also observed in Charcot-Marie-Tooth (CMT) disease, a genetically heterogeneous neurodegenerative disease caused by a mutation in FIG4 gene encoding PI(3,5)P_2_ 5-phosphatase, recognized as CMT-type 4J (CMT4J) [251]. CMT4J mouse models exhibit pale tremor (plt) phenotype and suffer from multi-organ disorder with peripheral neuronopathy, central nervous system degeneration and diluted pigmentation [251,252,253]. Importantly, fibroblasts, isolated from skin biopsies of CMT4J patients and CMT4J mice, display enlarged vacuoles that stain positive for LAMP-2 with concomitant impairments in the late endosome-lysosome pathway [251,252]. Furthermore, these enlarged vacuoles interfere with intracellular organelle trafficking and fail to properly incorporate cargo for processing within the late endosome-lysosome compartments [251].

### 10.3. Cardiovascular Diseases 

Defective lysosomal functioning is observed in cardiovascular diseases [254] and in multiple cells types given the hetero-cellularity of the heart. For e.g., the autophagy-lysosome pathway within macrophages plays an important role during atherosclerosis [254]. Macrophage-specific autophagy is important for lipid clearance [254], whereas autophagy-deficient macrophages remodel to be atherogenic in murine and human atherosclerotic lesions [254,255]. Upregulating the lysosome-autophagy pathway in macrophages either by overexpressing TFEB or treatment with the natural sugar trehalose, reverses autophagic dysfunction and attenuates macrophage apoptosis and pro-inflammatory IL-1β levels [254,255]. Moreover, macrophage-specific ATG5-null mice exhibit high atherosclerotic plaque formation and activation in a proatherogenic inflammasome [256]. In contrast, haploinsufficient Beclin-1 heterozygous-null mice do not display atherosclerotic phenotype, suggesting that the basal level of autophagy is crucial for atheroprotection [256]. Using a cell culture model of atherosclerosis, macrophages with high levels of oxidized-LDL and cholesterol crystals display an increase in lysosome pH, suppression of lysosomal proteolytic capacity and lysosomal dysfunction [257]. Macrophage-specific TFEB-overexpressing transgenic mice exhibit increased lysosome biogenesis and proteolytic capacity in macrophages with atherogenic lipids [257]. Furthermore, the lysosome-autophagy pathway is dysfunctional in mice with ischemia/reperfusion (IR) injury and in cardiac macrophages of humans with ischemic cardiomyopathy [258]. Transgenic mice with macrophage-specific TFEB overexpression demonstrate attenuated ventricular dysfunction and IL-1β secretion post IR injury, likely due to upregulation of lysosomal acid lipase [258]. 

Moreover, our laboratory also examined the importance of the lysosome-autophagy pathway in cardiomyocytes, specifically in a model of doxorubicin (DOX)-induced cardiotoxicity [259]. Indeed, we demonstrated that DOX negatively targets TFEB to suppress lysosome abundance and proteolytic functioning in cardiomyocytes, rendering cardiomyocytes susceptible to cell death [259]. Importantly, restoration of TFEB content using adenoviruses attenuated DOX-induced suppression in lysosome number, cathepsin activity and autophagy, and TFEB restoration also reduced DOX-induced increases in reactive oxygen species and cardiomyocyte injury [259]. Similarly, impairment in the lysosome-autophagy athway and decreases in lysosomal proteolytic activity is observed in the heart of type-1 diabetic Akita mice and mice fed high-fat high-sucrose diet for 16 weeks [260]. Notably, using an ex-vivo model of nutrient overload, we demonstrated that the saturated fatty acid palmitate decreases cellular TFEB levels resulting in decreased lysosomal content, suppressed lysosomal proteolytic capacity, impaired lysosome-autophagy function, and augmented cardiomyocyte injury [260]. These findings suggest that lysosome function is crucial for maintaining cardiomyocyte homeostasis and cardiomyocyte function.

Prior studies in humans have also highlighted the link between cardiac function and lysosome function in patients with Danon’s disease, which is characterized by an accumulation of intracytoplasmic vacuoles containing autophagic cargoes and glycogen in skeletal and cardiac muscle cells, resulting in X-linked vacuolar cardiomyopathy and myopathy [261,262,263]. LAMP2-deficient mice also displayed an accumulation of autophagic vacuoles in cardiomyocytes, along with reduced ejection fraction and reduced cardiac output compared to wild type mice [261,263]. These findings suggest that non-degraded autophagic vacuoles, due to deficiency of the lysosomal membrane protein LAMP2A, is causal for depressed cardiac function in patients with Danon’s disease [261]. The majority of previous research on lysosomal function, signaling and metabolism have been performed in non-cardiomyocytes, requiring a thorough investigation as to how intricacies of lysosomal signaling, transport, exocytosis and biogenesis pathways are involved in cardiac health and disease. 

## 11. Conclusion and Perspective

Classically, the lysosome is regarded as an organelle with distinct sorting, secretory and proteolytic function; however, newer data suggest that the lysosome is a metabolically active organelle dictating cellular homeostasis. In this review, we describe mechanisms of lysosomal sorting, biogenesis, functioning, positioning, and contact site formation. We also discuss how lysosome signaling events integrate with cellular metabolism specifically (1) discussing a modern view of lysosome sorting and positioning; (2) describing newer findings in processes targeting lysosomal size, positioning, contact site formation and biogenesis and how they are perturbed by metabolic changes; (3) providing an overview on lysosome ion mobilization and transport and its intricacies in relation to metabolic pathways and finally; (4) listing the challenges in translating these findings to treat diseases arising from aberrant lysosomal signaling, metabolism and function. Disruption or interference in lysosomal function, either by genetic mutations or loss of lysosomal protein function, is causative for lysosomal storage diseases, neurodegenerative diseases, cancer, cardiovascular diseases and metabolic disorders. Systemic approaches such as transcriptomics, metabolomics and proteomics and also imaging in diverse cells and organisms has added clarity to uncovering processes involved in maintaining lysosome function. However, we acknowledge that we have not discussed how hormones, vasculature, and neural inputs regulate lysosomal function. Furthermore, tissue specific transcriptional effectors of lysosome signaling are largely understudied. Despite the existing knowledge in understanding the multifaceted and heterogeneous role of the lysosome in regulating cellular homeostasis, there are numerous outstanding questions that still merit investigation. 

## Figures and Tables

**Figure 1 cells-09-01131-f001:**
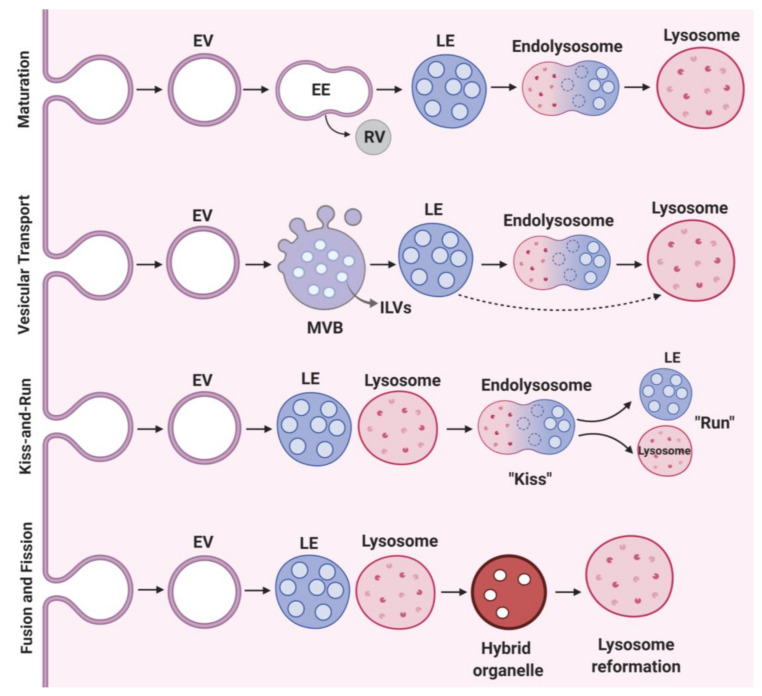
Molecular events in lysosome biogenesis. Maturation; this model of lysosome biogenesis describes the formation of EVs (endocytic vesicles) from the plasma membrane and their progressive maturation to late endosomes and subsequently to lysosomes. The cargo targeted for recycling is carried by the TGN derived RVs (recycling vesicles), whereas cargo required for lysosomal degradation is transported by the cargo vesicle through late endosomes. Vesicular transport; requires ECV/MVBs (endosomal carrier vesicle/multi-vesicular bodies) carrying ILVs (intraluminal vesicles), which mobilize cargo from early-to-late endosomes and then to lysosomes or mobilize cargo directly from the matured late endosomes to lysosomes. Kiss-and-Run; describes contact site formation between lysosomes and late endosomes (“kiss”) followed the by cargo transfer and ensuing dissociation (“run”) of late endosomes from lysosomes. Fusion and fission; events involve heterotypic fusion of late endosome-lysosome to form a hybrid organelle and subsequently lysosome reformation.

**Figure 2 cells-09-01131-f002:**
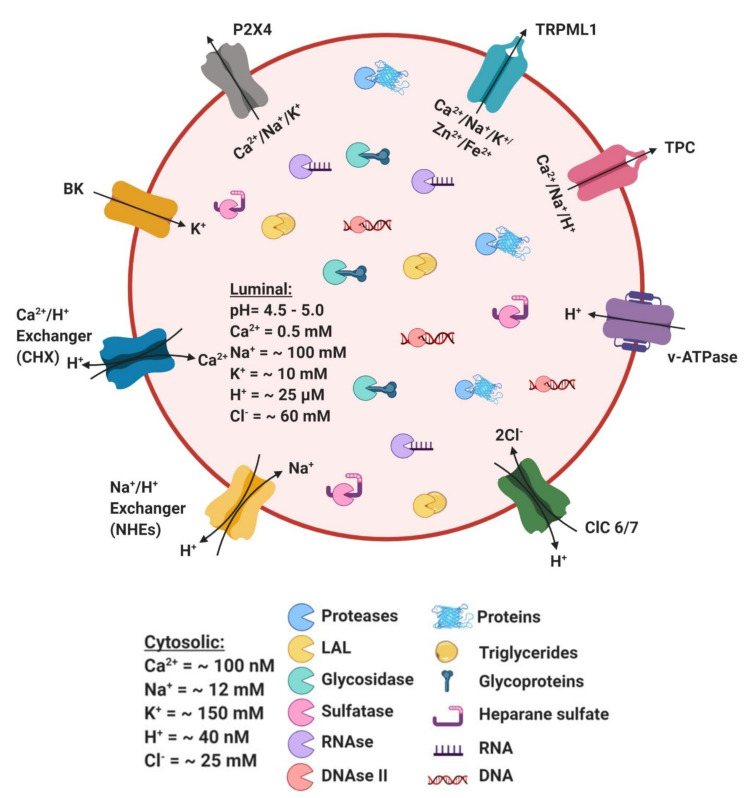
Lysosome ion channels and transporters. The lysosomal lumen contains a battery of soluble hydrolytic enzymes that degrade proteins (proteases), triglycerides (lysosomal acid lipase; LAL), glycoproteins (glycosidases), heparan sulphate (sulfatases), and nucleic acids (nucleases, RNAase and DNAse). Lysosomal ion channels and transporters include vATPase; vacuolar H^+^-ATPase (proton pump), chloride channel (ClC); ClC family that exchanges cytosolic Cl^−^ for lysosomal H^+^ (ClC-6 and ClC-7), K^+^ channels (BK). Non-selective cation channels include Transient Receptor Potential Cation Channel (TRPML); permeable to Ca^2+^, Na^+^, K^+^, Zn^2+^ and Fe^2+^, P2X4 channel; permeable to Ca^2+^, Na^+^ and K^+^ and two-pore channels (TPC); permeable to H^+^, Ca^2+^ and Na^+^. Ion transporters include Na^+^/H^+^ exchangers (NHEs) and Ca^2+^/H^+^ exchangers (CHX).

**Figure 3 cells-09-01131-f003:**
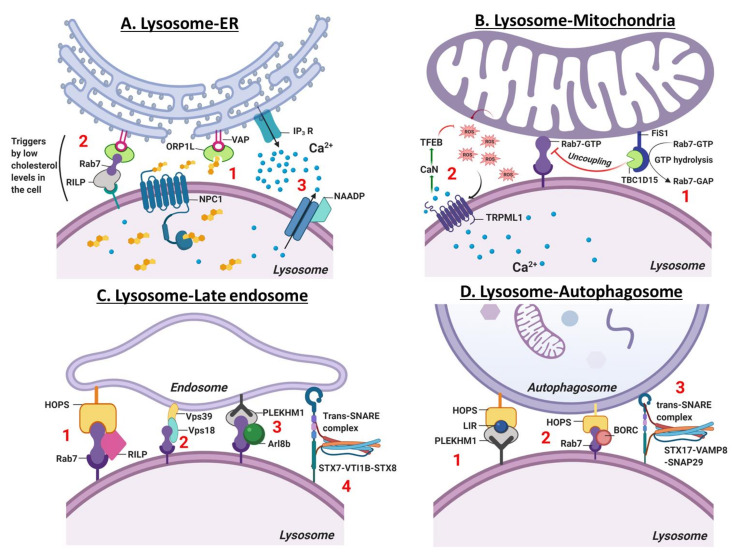
Organellar contact site and fusion with lysosome. (**A**). Lysosome-ER: Lysosome-ER contact site regulates ER Ca^2+^ release. (1) Lysosome-ER contact site shuttles free cholesterol from the lysosome to ER via concerted action of lysosomal membrane-localized protein, NPC1 and ER membrane proteins, VAPA (vesicle-associated membrane protein (VAMP)-associated ER protein A) and VAPB and a lipid-transfer protein, ORP1L (oxysterol-binding protein-related protein 1L). (2) Low cellular cholesterol level activates ORP1L, which then triggers the formation of a contact site between ER and lysosomal membrane. ER-resident protein VAP anchors with lysosomal Rab7-RILP (Rab7-interacting protein) to initiate the movement of lysosomes to plus-end or towards the cell periphery. (3) NAADP (nicotinic acid adenine dinucleotide phosphate), a Ca^2+^ mobilizing entity and IP3 receptor agonist mobilizes lysosomal Ca^2+^ and facilitates ER Ca^2+^ release into the cytosol. (**B**). Lysosome-Mitochondria: Lysosome-mitochondria contract site mediates the bidirectional regulation of lysosome and mitochondria dynamics. (1) Mitochondria forms a contact with the lysosome via lysosomal bound Rab7-GTP. This contact site is disrupted by mitochondrial fission 1 membrane protein, FIS1 and recruits TBC1D15 (TBC1 Domain Family Member 15) which hydrolyzes Rab7-GTP to Rab7-GAP. (2) Lysosomal mucolipin channel, TRPML1 (Transient Receptor Potential Cation Channel, Mucolipin subfamily) governs mitochondria homeostasis by sensing and scavenging mitochondrial ROS (reactive oxygen species). TRPML1 is activated by ROS that induces Ca^2+^ release from the lysosomal lumen and activates nuclear TFEB translocation in a calcineurin-dependent manner. TFEB induces lysosome biogenesis and autophagy to clear damaged mitochondria and scavenge ROS. (**C**). Lysosome-Late endosome: The fusion of lysosome and late endosome occurs via tethering processes, which involves; (1) coupling of small GTPase Rab7, Rab7-interacting protein (RIPL) and vacuole protein sorting (HOPS) complex, (2) interaction of Rab7, VPS18 and VPS39 (3) Rab7, Arf-like (Ar18b) and Rab7 effector PLEKHM1 (pleckstrin homology domain-containing protein family member 1 and (4) formation of Trans-SNARE assembly composed of syntaxin-7 (STX7, Qa-SNARE), VTI1B (VPS10 tail interactor-1B, Qb-SNARE) and syntaxin-8 (STX8, Qc-SNARE). (**D**). Lysosome-autophagosome: The fusion of lysosome and autophagosome occurs via (1) HOPS complex and Rab7 effector protein, PLEKHM1 which binds to HOP via LC3-interacting region (LIR) (2) BLOC-1 related complex (BORC1) that recruits Rab7 and HOPS for tethering and (3) enabling trans-SNARE complex formation of STX17-VAMP8-SNAP29 to induce autophagosome-lysosome fusion.

**Figure 4 cells-09-01131-f004:**
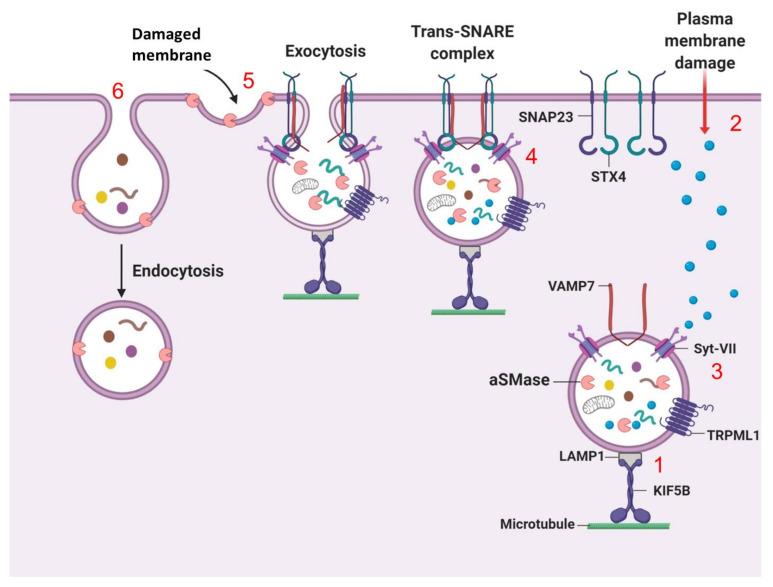
Lysosome exocytosis-dependent plasma membrane repair. Upon damage to the PM, (1) lysosomes are translocated to the cell periphery via cytoskeleton and motor protein, Kinesin 5 subfamily member KIF5B, which is bound to LAMP1 on the lysosomal membrane. (2) Injury to the PM can lead to an influx of Ca^2+^ in cells, which is then (3) sensed by Ca^2+^ sensor, Syt-VII (synaptotagmin-VII) on the lysosomal membrane and triggers a repair mechanism. (4) Subsequently, exocytosis is initiated by the lysosomal membrane localized VAMP7 (vesicle-associated membrane protein 7) that forms a Trans-SNARE complex with syntaxin-4 and SNAP23 (synaptosome-associated protein 23 kDa) on the PM. Trans-SNARE complex brings the lysosome in a close proximity to the PM to initiate the fusion of lysosome and PM. (5) The fusion of lysosome and PM triggers an efflux of lysosomal enzyme aSMase (sphingomyelinase), which is then retain on the PM to convert sphingomyelin into ceramide, (6) leading to an inward configuration of the PM to facilitate endocytosis-dependent removal and restoration of damaged PM.

**Figure 5 cells-09-01131-f005:**
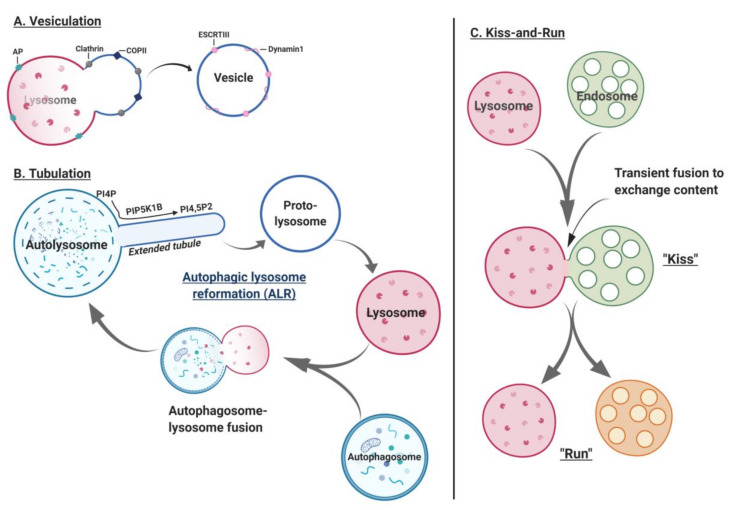
Lysosome fission. Lysosome fission involves, vesiculation process (**A**) is initiated by inner-coat adaptor protein-dependent cargo sorting, followed by membrane deformation mediated by clathrin and structural scaffold of COPII subcomplex. Subsequently, scission machinery including dynamin 1 and ESCRT-III catalyzes the membrane fission and dissociates vesicle from the lysosomal membrane. The process of tubulation (**B**) is best described by autophagic lysosome reformation (ALR), wherein proto-lysosomes derived from ALs, mature into functional lysosomes known as lysosome reformation. These matured lysosomes then subsequently fuse with autophagosomes to form ALs and initiate the cycle of lysosome reformation. The final event of lysosome fission is kiss-and-run (**C**), which involves the fusion of lysosomes and late endosomes. The transient interaction (“Kiss”) between lysosome and LEs allows the exchange of content, followed by separation of the lysosome (“Run”).

**Figure 6 cells-09-01131-f006:**
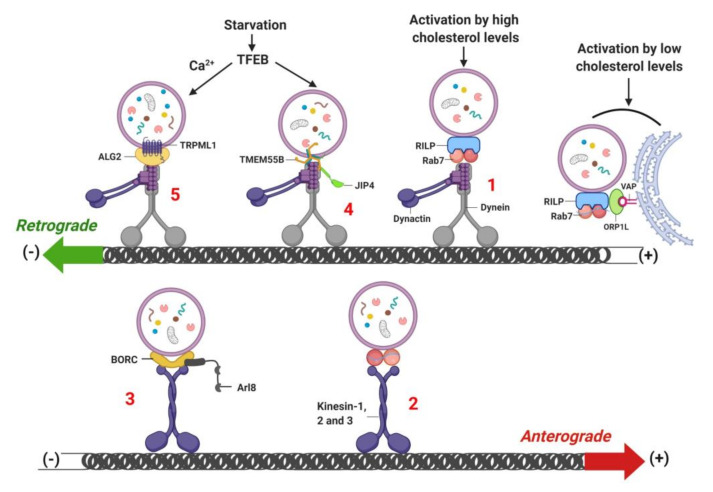
Retrograde- and anterograde-dependent lysosome mobilization. Lysosomes move from the cell periphery towards the perinuclear region in a retrograde manner. (1) The retrograde movement of the lysosome is initiated by RILP-Rab7-dependent recruitment and activation of dynein-dynactin, enabling the transport of lysosomes to minus-end direction. (2) While anterograde movement transport lysosomes from the perinuclear to the cell periphery, which requires kinesin-1, 2 and 3 motor proteins. (3) Anterograde movement of lysosomes also requires BLOC-1-related complex (BORC)-dependent recruitment of Arl8, which activates the kinesin proteins. (4) and (5) starvation-induced activation of TFEB increases the transcription of TRPML1 and TMEM55B to trigger lysosome mobilization in a retrograde manner via interaction with ALG2 (an EF-hand-containing protein, serve as lysosomal Ca^2+^ sensor) and JIP4 (C-Jun-amino-terminal kinase-interacting protein 4), respectively. Low cellular cholesterol levels restrict lysosomes at the plus end, which is mediated by the interaction between lysosomal RILP and Rab7 with VAP and ORP1L on ER membrane.

**Figure 7 cells-09-01131-f007:**
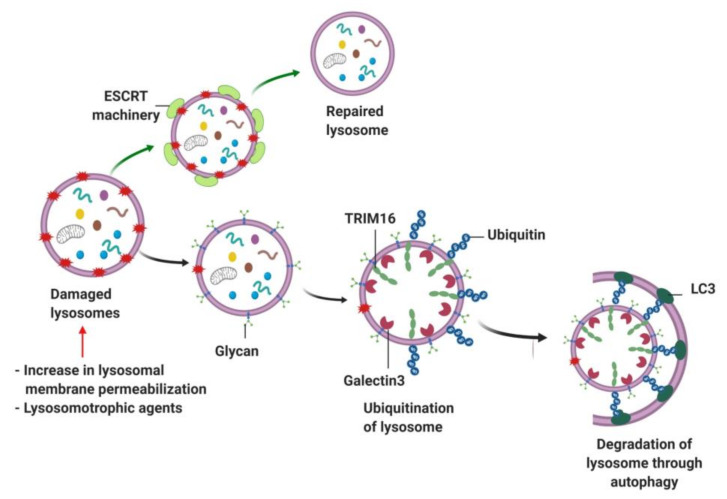
Lysosome repair by lysophagy. Damaged lysosomes can efflux luminal Ca^2+^ stores and activate ESCRT machinery component to repair lysosome by restoring lysosomal integrity. However, lysosomes are subjected to degradation when damaged. Severely damaged lysosomes can expose luminal glycan to the cytosol, which is sensed by galectin-3 to trigger lysosome ubiquitination via TRIM16. Following lysosome ubiquitination, lysophagy initiates lysosomal degradation.

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
