# Peer review of "Lysosomal Biology and Function: Modern View of Cellular Debris Bin"

_cells, 2020, doi:10.3390/cells9051131_

Round 1

Reviewer 1 Report

The paper by Trivedi and colleagues provides a detailed up to date review of the lysosome and associated functions. I did not find any factual errors, and the citations are appropriate. The article may benefit from editorial input; there is a lot of detailed information, but this is clarified by figures provided with the paper.

Author Response

The paper by Trivedi and colleagues provides a detailed up to date review of the lysosome and associated functions. I did not find any factual errors, and the citations are appropriate. The article may benefit from editorial input; there is a lot of detailed information, but this is clarified by figures provided with the paper.

Reviewer 1 did not have any concerns which needed to be address at the time of revision

Reviewer 2 Report

The manuscript by Trivedi et al, 2020 provides a detailed current understanding of “lysosomal biogenesis, signalling, metabolism and function­­”. In general, the review is well-written and the authors have included up-to-date list of discoveries made in the field and some interesting discussion throughout. The manuscript, however, can be further improved when the following comments are addressed.

Specific comments:

Please include references for some of the statements in the manuscript such as lines 67-72 or 125-141. In a few cases, the original research article(s) should be cited rather than review articles; For instance, in line 597 when the authors mentioned that ALR was discovered by Li Yu et al., 2007 or when mentioning specific findings in lines 839-842. Please go through the whole manuscript and make sure appropriate references included.

The reference of Figure 1 and Figure 2 to the information in the text is sometimes not appropriate. Please adjust this. It’s suggested that headings 7 and 8 become subheadings of the heading 6.

The sentences on lines 494-496 and 543-546 are not comprehensive. Please adjust these. Lines 414-417, “Indeed, to carry…. metabolism”, what is the subject of this sentence?

Lines 569-572, it is worth mentioning that knockout of Atg8s but not STX17 prevented autophagosome-lysosme fusion (DOI: 1083/jcb.201607039).

Lines 579-583, it’s said that “ limited data is available on lysosome fission event” (the authors should cite some papers as well). However, the next sentence stated as if this event has been well characterised: “For lysosome fission to occur, the lysosome must undergo a fusion process.” Is the latter statement has been proved by various studies or it is a theory/hypothesis? Please clarify this.

Line 589, which dynamin is involved?

Line 593, “as opposed to acute starvation” should be removed. Is that accurate to describe “autophagy” as “a proteolytic machinery” (line 776)? Please clarify this.

Lines 806-808, please cite some original articles to support the idea that “mitophagy … promoting the production of functional ATP-producing mitochondria”.

Line 835, is “autophagy” here “macroautophagy”? If so please adjust. Please make sure all abbreviations used in the manuscript are fully explained the first time they were mentioned.

Line 686, year should be added to the in-text reference. Line 706, it should be “autophagosome-lysosome fusion”.

Line 716-719, please use either lower case or upper case “dynein-dynactin” not both.

Line 793, this statement is not accurate. Please adjust it. Lines 795-800, the whole LC3/GABARAP family is involved not just LC3B. Could the authors please provide references to support that “p62 induces autophagosome-lysosome fusion?

Line 854-862: the way this paragraph ends makes this section unfinished. Please adjust the ending.

Typos: line 107 (“cargo” not “cardo”), line 226 (space between “enzymes” and “that”), line 282 (“RNase T2” instead of “RNASET2”), line 324 (“within” not “withing”), line 331 (“within” not “with”), line 453 (“this” not “which”), line 511 (“be” should be included before “crucial”), line 559 (“translocates” not “translocate”), line 562 (“HOPS” not “HOP”), line 617 (“prolonged” not “prolong”).

Author Response

Comments and Suggestions for Authors

The manuscript by Trivedi et al, 2020 provides a detailed current understanding of “lysosomal biogenesis, signalling, metabolism and function­­”. In general, the review is well-written and the authors have included up-to-date list of discoveries made in the field and some interesting discussion throughout. The manuscript, however, can be further improved when the following comments are addressed.

  1. Please include references for some of the statements in the manuscript such as lines 67-72 or 125-141.

We have incorporated the references as suggested by the reviewer.

  1. In a few cases, the original research article(s) should be cited rather than review articles;

     For instance, in line 597 when the authors mentioned that ALR was discovered by Li Yu et  

     al., 2007 or when mentioning specific findings in lines 839-842.

We have incorporated original references, as suggested by the reviewer.

  1. Please go through the whole manuscript and make sure appropriate references included. The reference of Figure 1 and Figure 2 to the information in the text is sometimes not appropriate. Please adjust this.

References are incorporated as suggested by the reviewers. Figure 1 and 2 are cited appropriately now.

  1. It’s suggested that headings 7 and 8 become subheadings of the heading 6.

Heading 7 is now a subheading of the heading 6

  1. The sentences on lines 494-496 and 543-546 are not comprehensive. Please adjust these.

This section is now revised as indicated in line 423-449.

The sentences between line 543-546 is now revised as indicated in line 465-476

  1. Lines 414-417, “Indeed, to carry…. metabolism”, what is the subject of this sentence?

Line 376-379 is now revised as suggested.

  1. Lines 569-572, it is worth mentioning that knockout of Atg8s but not STX17 prevented autophagosome-lysosome fusion (DOI: 1083/jcb.201607039).

We thank the author for pointing this out, and we now have included this data in line 491-493

  1. Lines 579-583, it’s said that “limited data is available on lysosome fission event” (the authors should cite some papers as well).

We have cited additional papers to support the findings in this section

  1. However, the next sentence stated as if this event has been well characterised: “For lysosome fission to occur, the lysosome must undergo a fusion process.” Is the latter statement has been proved by various studies or it is a theory/hypothesis? Please clarify this.

This section is revised and supported with additional references, detailed in line 536-546

  1. Line 589, which dynamin is involved?

Revised as indicated in line 536-546

  1. Line 593, “as opposed to acute starvation” should be removed. Is that accurate to describe “autophagy” as “a proteolytic machinery” (line 776)? Please clarify this.

Changes have been made as described in line 481-495

  1. Lines 806-808, please cite some original articles to support the idea that “mitophagy … promoting the production of functional ATP-producing mitochondria”.

We have now added relevant reference for this section as indicated in line 671-677

  1. Line 835, is “autophagy” here “macroautophagy”? If so please adjust.

This is now adjusted accordingly

  1. Please make sure all abbreviations used in the manuscript are fully explained the first time they were mentioned.

All the abbreviations are explained when described first time.

  1. Line 686, year should be added to the in-text reference. Line 706, it should be “autophagosome-lysosome fusion”.

Changes are incorporated as suggested.

  1. Line 716-719, please use either lower case or upper case “dynein-dynactin” not both. Line 793, this statement is not accurate. Please adjust it.

Reviewer changes are incorporated as suggested.

  1. Lines 795-800, the whole LC3/GABARAP family is involved not just LC3B. Could the authors please provide references to support that “p62 induces autophagosome-lysosome fusion?

Since autophagy is reviewed in detail elsewhere, we have condensed this section substantially.  

  1. Line 854-862: the way this paragraph ends makes this section unfinished. Please adjust the ending.

We have now adjusted the section as per the reviewer’s suggestion

  1. Typos: line 107 (“cargo” not “cardo”),
  2. line 226 (space between “enzymes” and “that”),
  3. line 282 (“RNase T2” instead of “RNASET2”),
  4. line 324 (“within” not “withing”),
  5. line 331 (“within” not “with”),
  6. line 453 (“this” not “which”), line 511 (“be” should be included before “crucial”),
  7. line 559 (“translocates” not “translocate”),
  8. line 562 (“HOPS” not “HOP”),
  9. line 617 (“prolonged” not “prolong”).

We have rectified the typographical and grammatical errors in response to comments 19-27.